# Availability, accessibility, and impact of social support on breast cancer treatment among breast cancer patients in Kumasi, Ghana: A qualitative study

Awolu Adam[1,2]*, Felix Koranteng[3]

1 Department of Family and Community Health, School of Public Health, University of Health and Allied Sciences, Ho, Ghana, 2 Center for Health Literacy and Rural Health Promotion, Accra, Ghana, 3 Department of Epidemiology and Biostatistics, School of Public Health, University of Health and Allied Sciences, Ho, Ghana

* aawolu@uhas.edu.gh

**Data Availability Statement:** The limited dataset has been uploaded to Zenodo and URL is https://zenodo.org/record/3551884#.XfL4cJNKjIU

**Funding:** The authors received no funding for this study.

## Abstract

### Background

Breast cancer is one of the top types of cancer affecting women both in the developed and developing countries. Breast cancer is a chronic and debilitating condition for anybody diagnosed of it and as well as their family. Social support has been shown to offset or moderate the impact of stress caused by the illness and other related negative outcomes.

### Objective

The objective of this study is to assess the availability, accessibility, and impact of social support on treatment for breast cancer patients at Komfo Anokye Teaching Hospital (KATH), Ashanti Region in Ghana.

### Materials and methods

A phenomenological study was employed. An in-depth interview guide was used to collect data on socio-demographic variables and social support availability and accessibility from 15 breast cancer patients. Thematic analysis was employed.

### Results

Majority of the women who participated in the study were postmenopausal women with an average age of 55 years. The study also revealed that all the participants in this study received one kind of support or another including informational, financial, emotional, and tangible support and reported varying positive impacts on their lives as a result of the support received. For those who received support, the prognosis and general quality of life appeared promising and well-adjusted than those who reported not having received any form of support.

**Competing interests:** The authors have declared that no competing interest exists.

## Conclusion

Social support is critical for the survival and quality of life of chronic disease patients including breast cancer patients who were the focus of this study. The availability and/or accessibility of social support or otherwise significantly determines the prognosis and quality of life of breast cancer patients. Healthcare professionals and family members or significant others are major players in organizing social support for chronic disease patients.

## Introduction

There is an increasing concern about the global burden of cancer, and breast cancer in particular, as millions of women around the world battle with breast cancer treatment and survival. Today, in addition to advanced countries, incidents and prevalence of breast cancer with associated mortality are on the rise even in developing countries. Reports from various sources indicate the increasing trend of breast cancer diagnosis despite the great efforts of advancement in technology to diagnose, image, and treat breast cancer. In 2018, estimates from the International Agency for Research on Cancer (IARC) showed that there were 2,088,849 new breast cancer diagnoses globally according to the data consolidated, and this constituted 11.6% of all incidents of cancers [1]. This was an increase of over 1.7 million new cases as compared to those reported in 2012 [2]. In terms of mortality, IARC reported that there were an estimated 626,679 breast cancer-related deaths in 2018 alone [1]. In 2018, 11.6% of all the reported cancer diagnosis were that of female breast cancer, which raises serious concerns over the efforts to prevent breast cancer globally [3].

Breast cancer, like all other types of cancer, is a global public problem and is prevalent across the world. Although the incidence and prevalence rates are different in different parts of the world, breast cancer incidence and prevalence have been observed and reported in both advanced and developing countries [4]. For instance, out of the over two million new diagnoses in 2018 stated above, 4.8% were from Asia, 5.8% from Africa, and the rest distributed around the Western countries. On the other hand, breast cancer-related mortality rates for the period in Asia and Africa were 57.4% and 7.2%, respectively [2].

The disease burden of breast cancer in terms of mortality is high and the rates of mortality vary around the world. Breast cancer-related mortality rates were estimated to range from 6 to 29 per 100 000, making breast cancer rank second in the cause of cancer related deaths as well as **morbidity**, and it is also the most frequent cause of cancer related deaths in women in both developing and developed regions [5]. The mortality rates are higher in less developed and resource-constrained countries as compared to the developed countries of the West. For example, only 32% of the women are still alive five years after a breast cancer diagnosis in sub-Sahara Africa as compared to 81% in the USA [6]. Again, Africa is reported to have the highest age-standardized mortality rates of breast cancer in the world even though the incidence rates are lower in Africa as compared to Western countries [7]. The increase in breast cancer incidents and mortality, especially in developing countries, has been attributed partly to the increasing population and the increased prevalence of risk factors associated with economic transition and the certain infectious agents of importance in cancer etiology [8].

Ghana has not been spared by the disease burden of breast cancer as breast cancer has had devastating effects on many families and communities and has been considered to be the commonest cancer in women in Ghana, in both incidence and mortality rates [9]. In 2014 alone, 5500 women died of cancer-related causes in Ghana and 18.5% of all cancer-related deaths in

women were due to breast cancer [9]. In that same year, 2,260 women were newly-diagnosed with breast cancer [9]. It is important to note that the above figures are reported figures and there may be more unreported and undiagnosed cases due to the resource limitations in terms of qualified oncologists and equipment to correctly diagnose. Radiology services such as mammograms and other forms of scans are not part of general health screening in the public healthcare system especially in rural communities in Ghana [9] and this makes it difficult for a quick and accurate diagnosis. Breast cancer is, therefore, a major killer of women in Ghana and a growing public health concern.

Cancer, and for that matter breast cancer diagnosis, is not just devastating news to those diagnosed but it is a life-threatening and changing event that can spell doom for many with negative outcomes. The survival rates depend on many factors which may include the time of diagnosis, nature and quality of treatment, and patient's personal factors. The cost of treatment is pretty high and inaccessible for many women in the developing countries. Besides that, the effects of treatment options such as chemotherapy and radiotherapy can result in serious adverse effects on the patients. How patients respond to both the diagnosis and treatment depends largely on the level of both professional and social support since breast cancer diagnosis can result in high levels of stress, anxiety, and depression. These, in turn, have the potential to negatively impact treatment outcomes for breast cancer patients as they go through treatment as the quality of life of breast cancer patients can be affected negatively.

Social support has been found to be correlated with positive treatment outcomes for many chronic conditions including breast cancer, and it significantly reduces the stress emanating from cancer diagnosis as well as improves emotional wellbeing [10]. Social support is described as support received in the form of information or a tangible item and emotional support; or the sources of support (example, family or friends) that enhance the recipients' self-esteem or provide stress-related interpersonal aid [10]. The results of several studies testify to the strong relationship between social support and the positive treatment outcomes for breast cancer patients. For instance, in a study to examine the effects of perceived social support from family and significant others for women undergoing breast reconstruction, researchers found that perceived social support was reported to reduce stress level and improved emotional wellbeing [11]. In a related study, social support in the form of sharing mutual experience among women in a breast cancer group resulted in not only reduced levels of stress but also an increased overall knowledge of breast cancer in addition to reducing negative thoughts and anxiety among women on breast cancer treatment options [12, 13]. In studying the results of changes in the quality and quantity of social support on breast cancer treatment, Fong and colleagues [14] found that social support reduces over the course of a year for breast cancer patients and concluded that the quality of social support was critical in maintaining stability for their emotional wellbeing. The critical need for social support in combatting stress, depression, anxiety, and improving the quality of life of breast cancer patients has been firmly established by the results of many other studies [15,16,17,18].

Lack of social support for chronic disease patients including breast cancer has been linked to poor emotional wellbeing, increased depressive symptoms, and poor quality of life [19]. Socially isolated women who lack access to care especially from social networks such as family and friends may develop an increased risk of mortality after breast cancer diagnosis [20] due to pain, depression, and poor emotional and mental wellbeing. Breast cancer patients with decreased social support report increased incidents of anxiety and depression [14].

With the disease burden of breast cancer increasing in Ghana and becoming a major public health concern, it is important that the availability, accessibility, and quality of social support for breast cancer patients in Ghana as well as the impact of the treatment outcomes on the patients and their families are examined. However, there is currently little research on social

support and its implications on breast cancer treatment in Ghana. The aim of this study was, therefore, to examine the availability, accessibility, and impact of social support for breast cancer patients receiving treatment at the Komfo Anokye Teaching Hospital (KAHT) in Kumasi, Ghana.

## Methods

A phenomenological study design was used to assess the availability of social support for breast cancer patients receiving breast cancer treatment at Komfo Anokye Teaching Hospital (KATH) in the Ashanti Region of Ghana. This is the type of qualitative study where 5–25 people are interviewed to collect data on common experiences about a particular phenomenon [21]. Although at different ages and diagnosed at different times in their lives, all the women who participated in this study had lived experiences of being diagnosed with breast cancer and were undergoing breast cancer treatment in the same facility. As such, the most appropriate design that we deemed fit to help elicit and describe the meanings the women gave to their lived experiences with regards to breast cancer and social support was the phenomenological design. This was because we sought to explore the availability of various forms of social support and describe the impact of the social support in enhancing the treatment outcomes for breast cancer patients.

### Sampling of participants

Breast cancer is a subject still not very much talked about openly in the traditional Ghanaian setup and many keep their diagnosis to themselves or to their families. Finding participants in the general population can be challenging. We decided to use convenience sampling, a non-probability sampling technique at the Breast Care Unit of Komfo Anokye Teaching Hospital (KATH) in Kumasi, Ghana. KATH is the second-largest tertiary hospital located in Kumasi in the Ashanti Region of Ghana. The choice of KATH was based on the availability of a Cancer Unit with a radiotherapy center and also because of the large number of breast cancer patients seeking treatment at the facility.

To conduct the sampling, the researchers were introduced to the patients by the administrator of the unit at the start of workday where breast cancer patients come for various treatments. We informed the patients about the study we intended to conduct and explained the purpose and details of the study to the participants including the objectives, procedures, time required to participate, and potential benefits and risks of participating in the study. At the time of the study, there were 29 women at the unit to receive various treatments and care. After being introduced to the women by the unit administrator we explained the purpose and steps involved in the study. We then invited the women to participate with emphasis that participation was completely voluntary. More women than we required for our sample were willing to participate in the study. Therefore, a simple random sampling was used to select seventeen (17) women after we assigned numbers to all the women who expressed interest to participate. Two women later dropped out of the study as they could not make time for the in-depth interviews.

Qualitative data were collected through in-depth interviews with 15 women who voluntarily consented to participate in the study in the breast cancer unit of KATH at the times convenient for them. The data collection lasted for six weeks from January 1st, 2018 to February 23rd, 2018 and the interview sessions were between 35 minutes to 45 minutes.

The inclusion criteria for the study were all women receiving breast cancer treatment at KATH at the time of the study who were at least 18 years of age, and who provided voluntary informed consent in written or verbal form to participate in the study. We excluded women

who were not receiving breast cancer treatment, females younger than 18 years, and women who did not voluntarily agree to participate. Permission was sought from each participant to tape-record the interviews. Family members who accompanied the participants witnessed the voluntary informed consent process. The 15 women who voluntarily participated in the study chose their own convenient days and times for the in-depth interviews and all agreed to interview at the Cancer Unit before they were seen by their treatment team. Thirteen women gave written consent. Two women gave verbal consent which was informed as all information with regards to the purpose and procedures involved in the study collection was explained to them in the Akan language. The women had no formal education and could neither read nor write. They expressed interest to be interviewed but stated they could not sign. They also did not want to thump print. Their family members witnessed their voluntary consent to be interviewed. A note was, therefore, made on the information sheets submitted to them by the interviewers and interview guides as documentary evidence that they verbally consented to participate. The consent was also tape recorded with permission from the participants.

The data collected was analyzed by transcribing the participants' responses and coding them using NVIVO 11 to generate themes and patterns. Similar responses were then grouped under the same theme and given codes. The participants' names were not used in the analysis and report writing. Some verbatim reporting was done in instances where the actual words of the participants were needed to make meaning or emphasize an important issue. Simple descriptive analysis was conducted on the respondents' demographics: including age, sex, marital status, and occupation, level of education, monthly income, and religion. Ethical review and approval of the study were done by the Ghana Health Service Ethics Review Committee with the approval number GHS-ERC:26/05/17.

## Results

### Socio-demographic characteristics

Demographic characteristics in a study such as this are crucial in understanding what the participants present and for this reason a number of important demographic variables were included in the interview guide. These variables included age, marital status, occupation, educational level attained by the participants, and duration of illness. The average age of the participants was 55 years, the minimum age was 35 years, and the highest age was 80 years. Nine (60%) out of the 15 participants were married, two (13.3%) were widowed, another two (13.3%) were divorced, one (6.7%) was separated, and one (6.7%) was single. Out of the 15 participants, five (33.3%) had no formal education, three (20%) attained tertiary education, and seven (46.7%) were Junior High School graduates. 13 (86.7%) were employed while two (13.3%) were not working at the time of the study. After examining the demographic characteristics above, the researchers wanted to find out how long the women had been living with their diagnosis. The results showed varying lengths of diagnosis. For instance, six (40%) of the participants reported they were diagnosed one year to the time of the interview and four had been ill for two years. Also, two women were ill for three years and three women for four years. The longest period reported was nine years by one woman. Another woman reported being diagnosed six years back. The above-described characteristics are presented in the Table 1.

### Availability and types of social support systems for breast cancer patients

A major objective of this study was to find out if breast cancer patients in this study received any kind of support as they were going through treatment. The participants were asked if they knew of or were given any kind of support from the hospital, from family and friends, church

**Table 1. Demographic characteristics of participants.**

| Variable | Frequency | Percent |
|---|---|---|
| **Age Group** | | |
| 31–40 | 2 | 13.3 |
| 40–50 | 2 | 13.3 |
| 51–60 | 9 | 60.0 |
| 61–70 | 1 | 6.7 |
| 71–80 | 1 | 6.7 |
| **Marital Status** | | |
| Single | 1 | 6.7 |
| Married | 9 | 60.0 |
| Divorced | 2 | 13.3 |
| Separated | 1 | 6.7 |
| Widow | 2 | 13.3 |
| **Religion** | | |
| Christian | 11 | 73.3 |
| Muslim | 4 | 26.7 |
| **Educational Attainment** | | |
| No formal education | 5 | 33.3 |
| Junior High School | 7 | 46.7 |
| Tertiary | 3 | 20.0 |
| **Employment Status** | | |
| Employed | 13 | 86.7 |
| Unemployed | 2 | 13.3 |
| **Duration of Illness (years)** | | |
| 0–1 | 6 | 40 |
| 2–3 | 4 | 26.7 |
| 4–5 | 3 | 20 |
| 6–7 | 1 | 6.7 |
| 8–9 | 1 | 6.7 |

members, and from other sources. The interviews revealed four types of support that the participants reported they received or were receiving at the time of the study, including informational, emotional, financial, and tangible support. For instance, 14 out of the 15 participants stated that the nurses at the hospital gave them important information about their condition and emotionally supported them. Seven of the participants also indicated that they got family support, which included mainly financial and emotional support. Again, some participants reported receiving support from their religious organizations in the form of material gives and home visits. So clearly, the results have shown that various types of social support were available and were accessed by the breast cancer patients in this study. In support of the various support types received, the following are verbatim statements from some of the participants.

**Informational support.** Almost all the 15 participants reported receiving information they considered useful especially from the health professionals they encounter in their treatment journey. The information received covered issues such as the type of food to eat, medication adherence, and self-care. In responding to a question as to whether they received information regarding breast cancer, below are some of quoted responses from participants.

*"As for that one, plenty plenty even when we came here they tell us some"*. [Participant 001, age 53].

"*They tell us not to be frightened or not to be afraid, not to be heartbroken, eat well and eat a lot of fruits*". [Participant 002, age, 54].

"*Yes, especially my doctor who first treated me. He explained it and made me understand that the injections and things that I will be injected will let me lose my hair and things but when I get better all the hair will come back. Truly truly it was so. I was on radiation machine and they counseled me too. . ...*". [Participant 003, age 58].

"*Yes, I have been educated here and even before I had it and came here, I learned about breast cancer. I encourage myself because my director counseled me a lot. His mother was a victim so he know how it is and encouraged me not to be sad*". [Participant 006, age 59].

"*Like the gentleman talking he shows us some leaves of a fruit the Ga people call 'aluguitugui'. He tells us to boil it and drink I have seen that it is very fine*".

[Participant 009, age 53].

"*Yes, the information from the nurses has really helped me*". [Participant 012, age 80]

"*Okay they tell us to eat fruit, like water melon, pineapple and pear and vegetables like 'konto-mire' and 'ayoyo' and banana and plenty fruits and we will be fine.*"

[Participant 015, age 50].

Concerning being given informational support on the treatment procedure and the conditions involved in the procedure, one of the participants indicated that she was made aware that some of the treatment procedures come with some adverse effects such as loss of hair and weight loss. This is what she said about informational support "*My doctor who first treated me explained and made me understand that the injections will let me lose my hair but when I get better all the hair will come and truly it was so. I was on radiation machine and they also counseled me, so for me I don't have a problem in this hospital*". [Participant 001, age 53].

Only one participant reported not getting valuable information from healthcare professionals. She has this to say. "*mmm if you come, they say go to the lab or go and take injection*". [Participant 007, age 55].

**Financial/material support.** In terms of receiving financial support, majority (9) of the participants reported receiving some financial support in paying for medication and taking care of other important things. On the other side, six participants reported they received no financial support and pay for treatments themselves. Below are some of the responses by those who received financial support:

"*In terms of financial support I used to get support from my coworkers. This person will send and this person will send but now I feel shy to tell them again. . ..*". [Participant 001, aged 53].

"*Yes, my brother is the paying for everything since I started treatment in 2008*". [Participant 003, 58]

"*It's God's grace, in fact. I'm on pension and I haven't received my pension pay yet and neither do I have monthly salary so it is friends and God's grace*". [Participant 006, aged 59].

"*People helped me. Family members did some and outsiders did some*". [Participant 007, age 55].

"*My son is in Abidjan. He pays some and my daughter used to take care of me but now her work has spoiled*". [Participant 012, age 80].

*"Apart from myself, in terms of financial support, I get support from my husband, my sister and my mother. They have been of great support."* [Participant 015, aged 50].

For those who did not receive financial support, below are some of the quotes.

*"No support. I am the who pays, my daughter goes with the money to buy the things like injections".* [Participant 005, age 70]

*"I am the only one who pays so when it happened like that, I wanted to build a house so I sold everything and use the money to help myself".* [Participant 009, age 53].

*"No ooh, I am the only who pays for my treatment".* [Participant 013, age 38]

**Food aid and home chores.** Besides financial support, other material support in the form of food aid and home support activities were noted by participants. The views expressed were diverse in nature. Some admitted having support from family in terms of food and household activities while others reported that they did receive food or house chores support at the time of the study. Below are some of their responses related to support in terms of food and household activities.

*"We are told to eat balanced diet every day, in the beginning I used to do as they told but now I don't get it like so I'm forced to eat whatever food I get and my daughter has been helping with the house chores."* [Participants 001, aged 53].

*"For home activities, my children are grown so I don't to do home activities. For food, yes I eat well".* [Participant 004, age 57].

*"It is my daughter, she has been helping with activities at home".* [Participant 007, age 55].

*"At the initial stage when I was discharged from the hospital with the stitches I wasn't doing anything but after the stitches were taken out I'm trying to do some petty activities in the house meanwhile I'm with my granddaughter barely two years old."* [Participant 006, aged, 59].

*"Yes, my grand child is the who helps me".* [Participant 012, age 80].

*"No I don't get help from anywhere, as for food no one helps me."* [-Participant 002, aged 54].

*"I do all my home activities by myself."* [Participant 005, aged 70].

*"I do home activities by myself and I eat well".* [Participant 013, age 38].

**Emotional support.** There was also the identification of emotional support as being crucial while undergoing treatment. Majority of the participants noted that family members, friends, and church members showed love and concern throughout their journey of fighting breast cancer as seen in some of the following interview quotes.

*"I live beautifully at home with my daughter and my husband especially my daughter she has been so helpful."* [Participant 001, aged 53].

*"Okay when I came for the surgery and left, church members and friends came to visit. My family keeps me company".* [Participant 009, age 53]

*"Yes, those of my friends who know come and visit. My church members visit me"*. [Participant 013, age 38].

*"Only a few people know I have cancer, but for my sister and my children who know I have cancer they show me love and concern." [Participant 015, aged 50].*

*"My daughter who is my last born and my close friend have shown me love as well as my director, I was in need and I ran to him and in fact he counseled me."* [Participant 006, aged 59].

*"I started the breast care November 2011, and then I came for surgery in 2012 and around 4years when they have added me to the first survivors when I started my 6 months reporting to support others."* [Participant 010, aged 59].

There were some who reported not having emotional support and two of them had the following to say.

*"No one comes to visit me. My church, no oo, they don't help me. They know but they only gave me small help, that is all"*. [Participant 002, age 54]

*"No I have no emotional support because I have not let people know I have the condition. I come to hospital here alone"*. [Participant 011, age 47]

**Spiritual support.** Concerning support from the participants' religious affiliations, majority of the participants stated that their church members supported and encouraged them, and showed them love. A few of them, however, stated that they did not receive any form of support from their church members and those who said they were Muslims also stated that they received no support from their Islamic family. Below are some of the responses.

*"I have even stopped going to church, I have joined a new church now, my previous church members didn't help me."* [Participant 003, aged 58].

*"It is like I go to church always but nobody really comes to visit me"*. [Participant 004, age 57]

*"Yes, my church members have been visiting me"*. [Participant 006, age 59]

*"I haven't made known to my church members I have cancer so they have no idea since they don't know they can't give me help."* [Participant 008, aged 38].

*"My church members helped me small'*. [Participant 009, age 53].

*"No support from my church at times my salary cannot pay so I plead with the pharmacist or if I can pay half when I receive the next salary I come to complete the payment."* [Participant 010, aged 59].

*"I'm a Muslim, they don't visit me."* [Participant 012, aged 80].

*"God will bless my church members, they have really helped me."* [Participant 005, aged 70].

*"My church people haven't helped me yet. I don't know if they will help me"*. [Participant 015, age 50]

From the above, while some attested that there were indeed support systems in the Komfo Anokye Teaching Hospital (KATH), others disagreed and mentioned that there was no

support. A woman aged 59 said, "*For support no I don't think so*". She added, *"I don't see the essence of the education being given by the health staff at the hospital, it does not make me feel better in anyway aside the treatment I'm given so I don't even listen when they are talking."* This is in agreement with an earlier research in which the researchers argued that an individual's ability to perceive the availability of support is highly likely to impact positively on health [22]. Thus, if one is unable to recognize that there is a support system available, it is likely he/she will not recognize the usefulness of the system.

Many of the different types of support that the participants received are categorized under the informational support of social support systems as defined by Kim and colleagues [10]. They also referred to social support as support received (including informative, emotional, or instrumental) or the sources of the support (example, family or friends) that enhance the recipients' self-esteem or provide stress-related interpersonal aid [10]. Education on the treatment procedure and the conditions associated with the procedure and education on the type of food to eat are all examples of informative support under the social support systems. Informational support included advice, suggestions, and other information that enhances the quality of life and provides a buffer against adverse life events.

### Impact of social support on breast cancer treatment

Aside from treatment and medication, social support according to research has shown to reduce anxiety, stress, fear, and other negative assumptions associated with being diagnosed with a disease condition, and for that matter breast cancer, and therefore, plays a critical role in the healing and treatment process of the disease condition.

In an attempt to find out if the availability of social support influenced the treatment of cancer in any way, the participants were asked some questions such as 'did the support you received help your treatment in any away?'

**Reduction in emotional distress.** According to two of the participants, the support helped reduce any form of discomforts and emotional pressures concerning their condition. Others noted that availability of social support had helped eliminate the fear associated with being diagnosed with breast cancer and all other false information linked to the treatment of the condition. Some of their views are summarized below.

On the basis of social support reducing discomforts associated with the condition, these are what two participants had to say;

*"Oh yes the education we are being given, like what the man is doing it has really helped me because I don't feel the way I used to feel when the condition first started".*

[Participant 014, aged 35].

*"I haven't experienced any discomfort ever since the sickness started, it's now that I have started experiencing discomfort."* [Participant 007, aged 46].

The responses given by the participants are in agreement with the arguments that positive social interactions have been shown to exert powerful beneficial effects on health outcomes and longevity [16]. This is also consistent with the argument that social support improves health by preventing or decreasing harmful physiological responses to stress, including cardiovascular reactivity to stress [22]. Concerning social support helping to eliminate the fear associated with being diagnosed of the condition and all other false information linked to the treatment of the condition, this is what one participant had to say.

"*I have encouraged myself and I'm not afraid anymore and I know I will be fine very soon.*" [Participant 002, aged 54].

Another participant mentioned;

"*I have been educated and before I even had the condition I have learnt a lot about it so I encouraged myself because my director counseled me a lot, his mother was a victim so he know how it is he encouraged me not be sad that kind of thing*". [Participant 006, aged 59].

It could be seen from the above narratives that the participants valued various social support as important source of knowledge and helping to deal with negative thoughts about breast cancer. This is consistent with the findings of other researchers in similar studies which mention that social support helps in decreasing anxiety, stress, and negative thoughts among patients of chronic diseases [15, 16, 17].

**Healthy nutritional choices.** With regards to social support influencing the lives and lifestyle of individuals, two participants agreed that indeed the various types of support they received have impacted their lives in a number of ways, including the choice of food and other aspects of their lives. Below are the summaries of what they said;

"*In terms of food it's normal, I have stopped eating certain foods like milk and meat. I have reduced my sugar in take. Right now my daughter is in SHS so when school opens and she leaves I suffer, this weekend liked this I really suffered.*" [Participant 009, aged 53].

### Effects of lack of social support systems on cancer patients

The availability of social support has proven to have a significant influence on not only the health of cancer patients but also on other aspects of life. This section sought to find out the possible effects the absence of these support systems can have on the cancer patients. Participants were interviewed to find out if the lack of the various types of support has in any way affected their livelihood. Some of the participants noted that lack of financial support forced them to sell their belongings, reduced their self-confidence and self-esteem, and some also said the lack of support has brought suffering on them.

**Financial and emotional burden.** Different types of difficulties were narrated by the participants who reported not receiving some forms of social support. Examples of the financial and emotional burden experienced are presented below. Regarding the lack of support resulting in suffering to their families, this is what one of them had to say;

"*My eldest son who was in the polytechnic to pursue his HND had to drop out of school, he was in school for only a year but because of a lot of money was used in treating the condition we made him stop the school and even me coming to the hospital today was very difficult, I had to borrow money, we even find it difficult to feed ourselves at home*". [Participant 001, aged 53].

Concerning selling belongings to be able to support themselves through the treatment of the condition due to lack of support, two participants noted that indeed they had to sell some belongings to support themselves. The following quotes summarize their responses;

". . ...*I cannot tell them again. It is like I am worrying them and they have wives and children. My husband is not working and it has brought suffering on me*". [Participant 001, age 53]

*". . ..with the series of test I did roughly I am heading towards about 6,000 Ghana cedis but now I am going to do the chemo, I don't know the amount I will be paying".* [Participant 006, age 59]

*"I wasted money and even with that I didn't know there was big loss afterwards so before they even gave me the date for the surgery I sold my belongings but later they checked and said I have to start the chemotherapy before so I came to start the chemotherapy."*

[Participant 007, aged 46].

*"I'm the only one who pays for my treatment so when it happened like that I had sell my building materials I wanted to build a house with to help myself."* [Participant 009, aged 53].

Regarding the lack of support leading to loss of self-esteem and self-confidence, one participant said:

*"I used to get support from my colleagues at work but now I feel shy to ask them again because I feel like I'm worrying them and besides they also have wives and children to take care of currently my husband is also not working and it has brought suffering on me."* [Participant 001, aged 53].

The above clearly documents critical information on social support availability and accessibility to chronic disease patients and particularly breast cancer patients at the KATH. Various social support types received and the positive and negative impact on the quality of life of breast cancer patients has been revealed in the study.

## Discussion

The importance of social support in enhancing positive treatment outcomes for people with chronic diseases and conditions is well-established [23,19, 11, 16]. This study sought to assess the availability and accessibility of social support for women receiving breast cancer treatment at Komfo Anokye Teaching Hospital in Kumasi, Ghana. There were a number of important lessons from the results presented above. First, the results of this study revealed the age groups of the women who participated in the study and a majority of them were postmenopausal women. Nine out of the 15 women in this study were between the ages 51 and 60 years, and one was between the age of 61–70 and another 71–80. Only four were in the age ranges from 31 to 50. This finding is inconsistent with the findings of similar research that mention the peak age of breast cancer incidents in Africans occurs in the premenopausal period, while it occurs in the postmenopausal period among the whites [24]. It is also inconsistent with the conclusion from a breast cancer screening and examination study in the same health facility which mentions that breast cancer affects young and premenopausal women and reported that the average age of 300 women who were presented with breast cancer was 49 [25].

We learned from this study that various forms of social support were available and were accessed by the breast cancer patients who participated in this study including emotional, financial, tangible, and informational support. The social support types reported here are in line with the types of social support described in the definitions and dimensions of social support [10] and those reported in a similar study elsewhere [15]. The most common support received by the participants in this study was informational support, which mainly came from the healthcare providers and professionals and was deemed by most of the participants to be helpful with regards to healthy nutrition, understanding breast cancer treatment procedures, and increasing their knowledge about breast cancer. Emotional and financial support were

also key support types received by the participants in this study as seven of the participants reported such support mainly from family members and significant others. This is consistent with findings in a similar study that emotional and tangible supports largely came from family members and significant others [15]. We, therefore, learned from this study that whilst informational support was provided by healthcare professionals, emotional and tangible support were mainly from family members. Hence, family members and healthcare professionals were key sources of social support for women with breast cancer in this study, and may be the case for other persons with different kinds of chronic disease [18].

The results also confirmed the findings of many previous studies on the positive effects of social support for chronic disease patients. For instance, majority of the participants in this study considered the informational support they received as beneficial in that it helped increase their knowledge about breast cancer. The benefits of healthcare professionals' informational support for improving knowledge of breast cancer among women receiving treatment was highlighted in previous studies [15,12, 13]. Apart from increasing knowledge, the participants who received informational, tangible, and emotional support reported becoming emotionally stable, and also less anxious and depressed. Again, this finding is consistent with other findings of increased emotional stability and decreased depressive symptoms [10, 22, 23].

From the above evidence of the positive effect of social support, it stands to reason that the absence of social support may result in negative prognosis and poor quality of life of breast cancer patients [14, 19, 20]. In this study, some participants expressed disappointments and loss of trust and confidence in their religious-affiliated bodies for not supporting them, at least emotionally, as they went through difficult treatment regimes. Persons suffering severe conditions and disability experience high levels of anxiety, depression, anger, and hopelessness; and in addition to the high cost of treatment, many cancer patients go through psycho-emotional turmoil in the recovery process [26]. Other studies have reported similar negative effects of lack of social support for breast cancer patients [19].

## Conclusion

Social support is critical for the survival and quality of life of chronic disease patients including breast cancer patients who were the focus of this study. While we cannot make a generalization based on the sample in this study, we found that the participants in this study valued various social support types described and presented in this study as important for them as they travelled through their treatment journeys. Informational, emotional, and tangible support are critical for enhancing positive treatment outcomes and the family members and health professionals are crucial sources of support. A system to support coordination of various types of support from the family and healthcare professionals in a teamwork fashion may yield better social support for breast cancer patients and bring positive treatment outcomes. More expanded studies on the impact of social support on breast cancer treatment and the quality of life of breast cancer patients are recommended.

## Acknowledgments

We wish to express our profound gratitude to Mr. Dzokoto Michael who assisted in data entry, and to all who have helped us in diverse ways in undertaking this research. We also acknowledge the breast cancer patients who volunteered and participated in the study. I would like to acknowledge the tremendous assistance of whom I deliberated issues about data analysis with. We finally acknowledge and thank Mr. James Nkrumah Attom and Ms. Danku Nadia for their time and support.

## Author Contributions

**Conceptualization:** Awolu Adam, Felix Koranteng.

**Formal analysis:** Awolu Adam, Felix Koranteng.

**Investigation:** Awolu Adam, Felix Koranteng.

**Methodology:** Awolu Adam, Felix Koranteng.

**Project administration:** Awolu Adam.

**Resources:** Awolu Adam, Felix Koranteng.

**Supervision:** Awolu Adam.

**Validation:** Awolu Adam.

**Writing – original draft:** Awolu Adam, Felix Koranteng.

**Writing – review & editing:** Awolu Adam.

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
