## [Decision Letter · Decision Letter 0]

16 Oct 2019

PONE-D-19-25605

Availability, Accessibility, and Impact of Social Support on Breast Cancer Treatment Among Breast Cancer Patients At Komfo Anokye Teaching Hospital in Ghana: A Qualitative Study.

PLOS ONE

Dear Dr. Adam,

Thank you for submitting your manuscript to PLOS ONE. After careful consideration, we feel that it has merit but does not fully meet PLOS ONE’s publication criteria as it currently stands. Therefore, we invite you to submit a revised version of the manuscript that addresses the points raised during the review process.

We would appreciate receiving your revised manuscript by Nov 30 2019 11:59PM. To enhance the reproducibility of your results, we recommend that if applicable you deposit your laboratory protocols in protocols.io, where a protocol can be assigned its own identifier (DOI) such that it can be cited independently in the future. For instructions see: http://journals.plos.org/plosone/s/submission-guidelines#loc-laboratory-protocols

We look forward to receiving your revised manuscript.

Kind regards,

Alvaro Galli

Academic Editor

PLOS ONE

Journal Requirements:

2. In your Methods section, please provide additional information about the participant recruitment method and the demographic details of your participants. Please ensure you have provided sufficient details to replicate the analyses such as: a) the recruitment date range (month and year), b) a description of any inclusion/exclusion criteria that were applied to participant recruitment, c) a description of how participants were recruited, and d) descriptions of where participants were recruited and where the research took place.

4.  We suggest you thoroughly copyedit your manuscript for language usage, spelling, and grammar. If you do not know anyone who can help you do this, you may wish to consider employing a professional scientific editing service.  

5. Please provide additional details regarding participant consent. In the ethics statement in the Methods and online submission information, please ensure that you have specified (1) whether consent was informed and (2) what type you obtained (for instance, written or verbal, and if verbal, how it was documented and witnessed). If your study included minors, state whether you obtained consent from parents or guardians. If the need for consent was waived, please ensure that you have discussed whether all data were fully anonymized before you accessed them and/or whether the IRB or ethics committee waived the requirement for informed consent.

Reviewers' comments:

Reviewer's Responses to Questions

**Comments to the Author**

1. Is the manuscript technically sound, and do the data support the conclusions?

Reviewer #1: Partly

2. Has the statistical analysis been performed appropriately and rigorously? 

Reviewer #1: N/A

3. Have the authors made all data underlying the findings in their manuscript fully available?

Reviewer #1: No

4. Is the manuscript presented in an intelligible fashion and written in standard English?

Reviewer #1: Yes

5. Review Comments to the Author

Reviewer #1: This manuscript is timely and well written. I would recommend a proofreading for grammatical and punctuation issues, i.e., p.4 line 74 has "as well as and it is." Very clear discussion about the cancer options for women in Ghana. The focus on social support is important, especially since there is a lack of awareness about how to manage breast cancer by community members.

Please provide more detail on why phenomenology was selected. It is appropriate but your audience may need more information about this method in qualitative research. The sampling is actually called "convenience" sampling, not convenient (p.7 line 146). You cannot generalize from this sample (see pp.17-18) about how the sample provides data inconsistent with a quantitative sample. Please revise that section.

In Table 1, please revise the duration of illness (years) elements. The categories need to be discrete, e.g., 0-1, 2-3, 4-5, etc. As they stand the categories are incorrect and presents more than 15 participants.

In the findings section, I would recommend headers about the types of support that emerged from the themes: Informational, Emotional, etc. This information will assist the reader in understanding which theme you are discussing. I might even discuss "food" as an additional theme that people discussed--it appears in the text when discussing healthcare providers and family.

6. PLOS authors have the option to publish the peer review history of their article (what does this mean?). If published, this will include your full peer review and any attached files.

Reviewer #1: No

---

## [Author Response · Author response to Decision Letter 0]

17 Dec 2019

The following were the comments from the reviewers and responses indicating how we addressed the comments.

 Review Comments to the Author

1. Reviewer #1: This manuscript is timely and well written. I would recommend a proofreading for grammatical and punctuation issues, i.e., p.4 line 74 has "as well as and it is." Very clear discussion about the cancer options for women in Ghana. The focus on social support is important, especially since there is a lack of awareness about how to manage breast cancer by community members.

Response: The word morbidity was left out that made the sentence looked awkward. The word is inserted and the sentence now reads correctly. This is on line 76 in bold. The manuscript was submitted to professional proof reading and editing firm True Editors. It was edited and we strongly feel the manuscript has improved remarkably.

2. Please provide more detail on why phenomenology was selected. It is appropriate but your audience may need more information about this method in qualitative research. 

Response: We have provided more detail on why phenomenology was chosen and the details can be found on pages 7 and lines 146-152. 

3. The sampling is actually called "convenience" sampling, not convenient (p.7 line 146). 

Response: Convenient has been replaced by convenience as the sampling technique and it is on line 154.

4. You cannot generalize from this sample (see pp.17-18) about how the sample provides data . inconsistent with a quantitative sample. Please revise that section.

Response: We are not generalizing from this as we have seen different or conflicting findings from other studies. In the conclusion we have made it clear we are not generalizing from this current study. The statement is found on lines 465-468.

5. In Table 1, please revise the duration of illness (years) elements. The categories need to be discrete, e.g., 0-1, 2-3, 4-5, etc. As they stand the categories are incorrect and presents more than 15 participants.

Response: The duration of illness in table 1 has been revised and the number of participants is corrected to 15. The table is on page 10.

6. In the findings section, I would recommend headers about the types of support that emerged from the themes: Informational, Emotional, etc. This information will assist the reader in understanding which theme you are discussing. I might even discuss "food" as an additional theme that people discussed--it appears in the text when discussing healthcare providers and family

Response: We have provided headers in the findings section as recommended and this can be found as sub-headings under types of social support. These sub-headings include informational, financial/material, food and home chores, emotional, and spiritual support. Under the impact of social support, we have provided reduction in emotional distress and health nutritional choices as sub-headings. Again, under effects of lack of social support, we have provided financial and emotional burden as a sub-heading.

---

## [Decision Letter · Decision Letter 1]

21 Jan 2020

PONE-D-19-25605R1

Availability, Accessibility, and Impact of Social Support on Breast Cancer Treatment Among Breast Cancer Patients At Komfo Anokye Teaching Hospital in Ghana: A Qualitative Study.

PLOS ONE

Dear Dr Adam,

Thank you for submitting your manuscript to PLOS ONE. After careful consideration, we feel that it has merit but does not fully meet PLOS ONE’s publication criteria as it currently stands. Therefore, we invite you to submit a revised version of the manuscript that addresses the points raised during the review process.

We would appreciate receiving your revised manuscript by Mar 06 2020 11:59PM. To enhance the reproducibility of your results, we recommend that if applicable you deposit your laboratory protocols in protocols.io, where a protocol can be assigned its own identifier (DOI) such that it can be cited independently in the future. For instructions see: http://journals.plos.org/plosone/s/submission-guidelines#loc-laboratory-protocols

We look forward to receiving your revised manuscript.

Kind regards,

Alvaro Galli

Academic Editor

PLOS ONE

Reviewers' comments:

Reviewer's Responses to Questions

7. PLOS authors have the option to publish the peer review history of their article (what does this mean?). If published, this will include your full peer review and any attached files.

Reviewer #2: No

Reviewer #3: No

Reviewer #4: No

**Comments to the Author**

1. If the authors have adequately addressed your comments raised in a previous round of review and you feel that this manuscript is now acceptable for publication, you may indicate that here to bypass the “Comments to the Author” section, enter your conflict of interest statement in the “Confidential to Editor” section, and submit your "Accept" recommendation.

Reviewer #2: (No Response)

Reviewer #4: All comments have been addressed

2. Is the manuscript technically sound, and do the data support the conclusions?

Reviewer #2: Partly

Reviewer #4: Yes

3. Has the statistical analysis been performed appropriately and rigorously? 

Reviewer #2: N/A

Reviewer #4: Yes

4. Have the authors made all data underlying the findings in their manuscript fully available?

Reviewer #2: Yes

Reviewer #4: Yes

5. Is the manuscript presented in an intelligible fashion and written in standard English?

Reviewer #2: Yes

Reviewer #4: Yes

6. Review Comments to the Author

Reviewer #2: Comments to the Authors:

1. There are still some spelling and grammatical errors.

2. The organization of topics could be improved. "Food aid" is addressed in one section, and "Nutritional choices" in another, using the same quote. A quote about being a survivor is listed under "Nutritional choices." A paragraph about informational support is included under the heading of "spiritual support."

3. The conclusion continues to generalize from this study that social support determines the prognosis and quality of life of breast cancer patients. This study is not designed to answer that question.

4. In the introduction section, the difficulty in a quick and accurate diagnosis of breast cancer in Ghana is attributed to the lack of radiation therapy centers. It is not the function of radiation therapy centers to diagnose breast cancer, but rather to treat it. Diagnosis requires radiology services.

2. There is no clear description of how the sample was chosen. In one place it is described as a convenience sample. In another the authors state that all of the study participants had some experience with social support, indicating that this was an inclusion criterion. If so, that would introduce a selection bias. It also contradicts the statement that "A major objective of this study was to find out if breast cancer patients in this study received any kind of support..."It is stated that all of the participants were undergoing breast cancer treatment, yet many of them were several years from their diagnosis. A detailed and clear description of how the sample was chosen is needed.

3. There is very little support in the quotations cited for the statement that "participants not only gained more knowledge about breast cancer but were relieved of negative thoughts and were full of hope." There is insufficient data to make that claim.

Reviewer #4: This is a well written study and needed study to assess the social needs of breast cancer patients in a developing country like Ghana. As demonstrated, the women diagnosed with breast cancer desire social support and it appears to be a factor in their survivorship. Authors have addressed the reviewers comments.

---

## [Author Response · Author response to Decision Letter 1]

7 Feb 2020

We have attached our responses to the comments from reviewers. We are grateful for the kindid and comprehensive review conducted.

---

## [Decision Letter · Decision Letter 2]

27 Feb 2020

PONE-D-19-25605R2

Availability, Accessibility, and Impact of Social Support on Breast Cancer Treatment Among Breast Cancer Patients in Kumasi, Ghana: A Qualitative Study.

PLOS ONE

Dear Dr. Adam,

Thank you for submitting your manuscript to PLOS ONE. After careful consideration, we feel that it has merit but does not fully meet PLOS ONE’s publication criteria as it currently stands. Therefore, we invite you to submit a revised version of the manuscript that addresses the points raised during the review process.

We would appreciate receiving your revised manuscript by Apr 12 2020 11:59PM. To enhance the reproducibility of your results, we recommend that if applicable you deposit your laboratory protocols in protocols.io, where a protocol can be assigned its own identifier (DOI) such that it can be cited independently in the future. For instructions see: http://journals.plos.org/plosone/s/submission-guidelines#loc-laboratory-protocols

We look forward to receiving your revised manuscript.

Kind regards,

Alvaro Galli

Academic Editor

PLOS ONE

Reviewers' comments:

Reviewer's Responses to Questions

**Comments to the Author**

1. If the authors have adequately addressed your comments raised in a previous round of review and you feel that this manuscript is now acceptable for publication, you may indicate that here to bypass the “Comments to the Author” section, enter your conflict of interest statement in the “Confidential to Editor” section, and submit your "Accept" recommendation.

Reviewer #2: (No Response)

Reviewer #4: All comments have been addressed

2. Is the manuscript technically sound, and do the data support the conclusions?

Reviewer #2: Partly

Reviewer #4: Yes

3. Has the statistical analysis been performed appropriately and rigorously? 

Reviewer #2: N/A

Reviewer #4: Yes

4. Have the authors made all data underlying the findings in their manuscript fully available?

Reviewer #2: Yes

Reviewer #4: Yes

5. Is the manuscript presented in an intelligible fashion and written in standard English?

Reviewer #2: Yes

Reviewer #4: Yes

6. Review Comments to the Author

Reviewer #2: Although the sample is very small, it would be useful to see how many received the various types of social support instead of just a few quotes. There are no quotes from a third of the sample, and many of the quotes come from the same person, so it is hard to get a sense of how widespread the types of support are. The discussion continues to link the availability of social support in this sample to treatment outcome, which is not supported by either this data set or by the literature cited.

Reviewer #4: Authors have addressed the reviewers comments. The concerns regarding grammar and/or spelling were also addressed appropriately.

7. PLOS authors have the option to publish the peer review history of their article (what does this mean?). If published, this will include your full peer review and any attached files.

Reviewer #2: No

Reviewer #4: No

---

## [Author Response · Author response to Decision Letter 2]

17 Mar 2020

The comments have been addressed. Quotes from more participants added. Conclusion revised.

---

## [Decision Letter · Decision Letter 3]

31 Mar 2020

Availability, Accessibility, and Impact of Social Support on Breast Cancer Treatment Among Breast Cancer Patients in Kumasi, Ghana: A Qualitative Study.

PONE-D-19-25605R3

Dear Dr. Adam,

We are pleased to inform you that your manuscript has been judged scientifically suitable for publication and will be formally accepted for publication once it complies with all outstanding technical requirements.

With kind regards,

Alvaro Galli

Academic Editor

PLOS ONE

Additional Editor Comments (optional):

Reviewers' comments:

Reviewer's Responses to Questions

**Comments to the Author**

1. If the authors have adequately addressed your comments raised in a previous round of review and you feel that this manuscript is now acceptable for publication, you may indicate that here to bypass the “Comments to the Author” section, enter your conflict of interest statement in the “Confidential to Editor” section, and submit your "Accept" recommendation.

Reviewer #4: All comments have been addressed

2. Is the manuscript technically sound, and do the data support the conclusions?

Reviewer #4: Yes

3. Has the statistical analysis been performed appropriately and rigorously? 

Reviewer #4: Yes

4. Have the authors made all data underlying the findings in their manuscript fully available?

Reviewer #4: Yes

5. Is the manuscript presented in an intelligible fashion and written in standard English?

Reviewer #4: Yes

6. Review Comments to the Author

Reviewer #4: The authors have addressed reviewers' comments and amended the manuscript to reflect these changes.

7. PLOS authors have the option to publish the peer review history of their article (what does this mean?). If published, this will include your full peer review and any attached files.

Reviewer #4: No

---

## [Editor Report · Acceptance letter]

3 Apr 2020

PONE-D-19-25605R3 

Availability, Accessibility, and Impact of Social Support on Breast Cancer Treatment Among Breast Cancer Patients in Kumasi, Ghana: A Qualitative Study. 

Dear Dr. ADAM:

I am pleased to inform you that your manuscript has been deemed suitable for publication in PLOS ONE. Congratulations! Your manuscript is now with our production department. 

With kind regards,

on behalf of

Dr. Alvaro Galli 

Academic Editor

PLOS ONE